# Surface Modification of Electrospun Bioresorbable and Biostable Scaffolds by Pulsed DC Magnetron Sputtering of Titanium for Gingival Tissue Regeneration

**DOI:** 10.3390/polym14224922

**Published:** 2022-11-15

**Authors:** Arsalan D. Badaraev, Dmitrii V. Sidelev, Anna I. Kozelskaya, Evgeny N. Bolbasov, Tuan-Hoang Tran, Alexey V. Nashchekin, Anna B. Malashicheva, Sven Rutkowski, Sergei I. Tverdokhlebov

**Affiliations:** 1Weinberg Research Center, School of Nuclear Science & Engineering, National Research Tomsk Polytechnic University, 30, Lenin Avenue, 634050 Tomsk, Russia; 2Ioffe Institute, 26, Polytekhnicheskaya Street, 194021 St. Petersburg, Russia; 3Institute of Cytology RAS, 4, Tikhoretsky Avenue, 194064 St. Petersburg, Russia

**Keywords:** electrospinning, pulsed DC magnetron sputtering, titanium coating, plasma surface modification, polymer scaffolds, gingival fibroblasts, PLGA, VDF-TeFE

## Abstract

In this study, polymer scaffolds were fabricated from biodegradable poly(lactide-co-glycolide) (PLGA) and from non-biodegradable vinylidene fluoride-tetrafluoroethylene (VDF-TeFE) by electrospinning. These polymer scaffolds were subsequently surface-modified by sputtering titanium targets in an argon atmosphere. Direct current pulsed magnetron sputtering was applied to prevent a significant influence of discharge plasma on the morphology and mechanical properties of the nonwoven polymer scaffolds. The scaffolds with initially hydrophobic properties show higher hydrophilicity and absorbing properties after surface modification with titanium. The surface modification by titanium significantly increases the cell adhesion of both the biodegradable and the non-biodegradable scaffolds. Immunocytochemistry investigations of human gingival fibroblast cells on the surface-modified scaffolds indicate that a PLGA scaffold exhibits higher cell adhesion than a VDF-TeFE scaffold.

## 1. Introduction

The gingiva is a soft tissue of the oral mucosa that covers the alveolar bone and the roots of the teeth. The gingival tissue prevents the penetration of food residues into the tooth roots and also protects them from mechanical damage. Gingival recession is a very common disease in dentistry that exposes the roots of the teeth [1]. This can contribute to serious complications that can lead to tooth loss. The “gold standard” of gingival recession treatment is surgery with an autogenous subepithelial connective tissue graft [2]. The general treatment method is based on the removal of the affected part of the patient’s palate, which increases the risk of postoperative complications [3]. One of the most promising solutions to this problem is the use of polymer scaffolds, which promote the regeneration of gingival tissue [4,5]. The use of cell-friendly scaffolds reduces the trauma of a surgical procedure in the treatment of gingival recession.

Vinylidene fluoride-tetrafluoroethylene (VDF-TeFE) is a non-biodegradable, thermoplastic fluoropolymer with high hydrophobic, mechanical and piezoelectric properties [6,7]. Polymers containing vinylidene fluoride (VDF) and with piezoelectric properties are actively used in the field of tissue engineering [8,9]. Piezoelectric properties of polymers containing VDF have a positive effect on cell proliferation and differentiation [10,11]. Previously, we have shown that VDF-TeFE is a promising material in the treatment of mucosal defects in rats [12].

Despite the advantages of VDF-TeFE, a significant disadvantage of this polymer is its non-biodegradability, which leads to the need to remove it in an additional operation. Various biodegradable scaffolds are currently being produced that solve this problem due to their ability to degrade in the human body. The most common synthetic polymers used as scaffolds are the following polyesters: polylactide (PLA) [13,14], poly(lactide-co-glycolide) (PLGA) [15] and poly(ε-caprolactone) (PCL) [13,16]. Among various aliphatic polyesters, the copolymer PLGA is the most promising. In addition to biodegradability, biocompatibility and good mechanical properties, PLGA has a controllable degradation rate [17]. Due to their properties, PLGA scaffolds are actively used for gingival tissue regeneration [18,19].

One of the best studied and most effective methods for the fabrication of polymer scaffolds for use in tissue engineering is electrospinning [20,21]. This method can be used to obtain highly porous, mechanically stable, non-woven polymer structures with an extremely high surface-to-volume ratio [22]. Fiber diameters of polymer materials obtained by the electrospinning method are in the range of 3 nm to 5 µm [23]. Thereby, it is possible to produce polymer structures that imitate an extracellular matrix topology [24]. The main function of these polymer structures is to be a spatial-physical scaffold for living cells and to enhance their proliferation, migration and differentiation.

Despite the advantages, PLGA and VDF-TeFE polymers fabricated by electrospinning have hydrophobic properties [12,25]. A high hydrophobicity reduces the adhesion of cells to the scaffold surface, which has a negative effect on their proliferation. The hydrophobic properties and low surface energy of such polymers limit their medical use in the field of tissue regeneration.

The deposition of titanium on the surface of polymer materials improves their wetting properties [26,27]. Titanium is often used as a material for dental [28] and orthopedic [29] implants due to its high corrosion and wear resistance, hardness and biocompatibility. Magnetron sputtering is used to produce high-purity thin films and coatings with a thickness of several nanometers to microns. Direct current (DC) magnetron sputtering in a pulsed mode is more suitable to modify temperature-sensitive materials compared to continuous DC magnetron sputtering. This is caused by lower discharge current values for pulsed DC sputtering compared to non-pulsed DC sputtering. Pulsed DC magnetron sputtering enables the modification of polymer scaffolds without causing damage, which can be induced by temperatures above the glass-transition temperature or the melting point of polymers. El-Hossary et al. reported that titanium-containing thin films fabricated by pulsed DC magnetron sputtering show good tribological properties and a high biocompatibility with osteoblast cells [30]. He et al. describe that thin films of Ti and TiO_2_ fabricated by pulsed DC magnetron sputtering display no toxicity to osteoblast cells and have good cell adhesion and proliferation [31].

The plasma surface modification of PLGA and VDF-TeFE scaffolds by pulsed DC magnetron sputtering will improve the biological and wetting properties of scaffolds without changing their initial morphology, structure and mechanical characteristics. These improved surface properties of titanium-modified PLGA and VDF-TeFE scaffolds make them promising candidates for applications in the field of gingival tissue regeneration. Therefore, the aim of this work is to investigate titanium-coated electrospun PLGA and VDF-TeFE scaffolds to evaluate their potential use as materials for the regeneration of gingival tissue.

## 2. Materials and Methods

### 2.1. Scaffold Fabrication

A 4 wt% PLGA solution was prepared from poly(lactic-co-glycolic acid) granules (PLGA, 85/15, Corbion Purac, Amsterdam, The Netherlands) in hexafluoroisopropanol ((CF3)2CHOH, P&M Invest, Moscow, Russia) and a 5 wt% VDF-TeFE solution was prepared from vinylidene fluoride-tetrafluoroethylene granules (VDF-TeFE, Galopolymer, Moscow, Russia) in acetone (C_3_H_6_O, Ekos-1, Moscow, Russia). After electrospinning, all fabricated scaffolds were stored separately for one week in a dark, dry place at room temperature in order to let the solvents evaporate gradually from the samples. The polymer scaffolds were fabricated by the electrospinning method on a NANON-01A setup (MECC Co., Fukuoka, Japan), equipped with a cylindrical collector with a length of 200 mm and a diameter of 100 mm, which rotated at a speed of 200 rpm. For the fabrication of PLGA scaffolds, the following electrospinning parameters were applied: voltage: 22 kV, distance between the nozzle and the collector: 150 mm, solution flow rate: 4 mL/h, syringe volume: 10 mL, spinneret width: 200 mm, spinneret speed: 10 mm/s, a 20 G needle. The VDF-TeFE scaffolds were fabricated under the following electrospinning parameters: voltage: 20 kV, distance between nozzle and collector: 150 mm, solution flow rate: 6 mL/h, syringe volume: 10 mL, spinneret width: 200 mm, spinneret speed: 10 mm/s, a 22 G needle. The morphology of the scaffolds fabricated differs significantly since the used parameters of electrospinning cause these differences for polymers, and mechanically stable VDF-TeFE and PLGA scaffolds can be fabricated with these parameters. To remove the excess solvent, the electrospun scaffold samples were dried for 48 h in a drying oven at room temperature and a pressure of 0.5 Pa.

### 2.2. Surface Modification Procedure

Polymer scaffolds were modified by titanium (Ti-modified) using an ion-plasma setup equipped with a magnetron sputtering system [32]. For this modification, an APEL-M-5PDC power supply (Applied Electronics, Tomsk, Russia) working in unipolar mode (according to the reference [33]) with a frequency of 100 kHz and a duty cycle of 70% was connected to a magnetron with a titanium (99.95%) target. The polymer scaffolds were modified under the following parameters: discharge power: 750 W, voltage: 500 V, current: 1.5 A, modification time: 35 min, working pressure in the chamber: 0.3 Pa, working gas: argon (Ar, 99.99%), target shape: a disk with a diameter of 90 mm and a thickness of 8 mm. The scaffolds were fixed on substrate holders that were axially rotated during plasma modification (the radius of axial rotation was 150 mm). The magnetron sputtering time for the surface modification of the scaffolds was 35 min, with a break of 20 min. This break was performed after 17.5 min of magnetron sputtering, which continued for another 17.5 min after the break to complete the surface modification of the scaffolds. This break was necessary to prevent the polymer samples from damage caused by the high temperatures that occur during the plasma modification process. Additionally, the distance from the Ti target of the magnetron sputtering system to the scaffolds was selected as 150 mm to decrease the influence of the charged particles on the scaffolds. The deposition rate of 5.7 ± 0.3 nm/min was calculated using a quartz thickness gauge (Micron-5, Izovac, Minsk, Belarus). Based on the determined deposition rate and the modification time of 35 min, the coating thickness is about 200 ± 10 nm. It is important to note that the thickness of the coatings was measured on a smooth substrate, not on the surface of the polymer scaffolds.

These parameters were also applied in another study in which the influence of surface modification with copper and titanium coatings on the morphology and the mechanical and biological properties of nonwoven polymer scaffolds made of VDF-TeFE was investigated [34].

In the following, the unmodified PLGA and VDF-TeFE scaffolds are abbreviated as: u-PLGA and u-VDF-TeFE. The titanium surface-modified scaffolds are abbreviated as: Ti-PLGA and Ti-VDF-TeFE. The experimental scheme of the fabrication via electrospinning of the investigated polymer scaffolds and the surface modification by magnetron sputtering are displayed in Figure 1.

### 2.3. Characterization of the Scaffolds

*Scaffold Surface Morphology.* A scanning electron microscope (SEM), the JCM-6000 Plus (Jeol, Akishima, Japan), was used to investigate the morphology of the unmodified and plasma-modified PLGA and VDF-TeFE scaffold surfaces. The micrographs were obtained at a magnification of ×1000 at an accelerating voltage of 15 kV and under high vacuum conditions. A thin film of gold was deposited on the surface of the PLGA scaffolds using a SmartCoater device (Jeol, Akishima, Japan) to provide charge dissipation. The fiber diameter histograms were obtained from the SEM micrographs using the DiameterJ v1.018 (National Institute of Standards and Technology, Gaithersburg, MD, USA) plug-in in the ImageJ 1.48 software (National Institute of Health, Washington, DC, USA).

*Elemental and Chemical Composition of the Scaffold Surfaces.* Elemental analysis of polymer scaffolds by energy dispersive X-ray spectroscopy (EDX) was performed on a JCM-6000 Plus instrument equipped with an EDX analyzer (Jeol, Akishima, Japan). EDX spectra of the scaffolds were performed at a magnification of ×500 at an accelerating voltage of 15 kV, with an average EDX spectra acquisition time of 120 s. The elemental composition raw data were evaluated according to the ZAF correction (abbreviation for: Z—atomic number, A—absorption effect and F—fluorescence excitation effect) to obtain quantitative elemental composition results of the scaffold surfaces.

The chemical composition study was carried out using an X-ray photoelectron spectroscope (XPS, Escalab 250Xi, Thermo Fisher Scientific Inc., Waltham, MA, USA). A monochromatic Al Kα radiation source with a photon energy of 1486.6 eV was used for the XPS investigations. To obtain a survey and an element core-level spectrum pass, energies of 100 eV and 50 eV were applied, respectively. The spot size of the X-ray beam was 650 µm. The atomic concentration measurements and deconvolution of the XPS spectra were determined with the CasaXPS Version 2.3.25PR1.0 software (Thermo Fisher Scientific Inc., Waltham, MA, USA). Ion beam treatment of polymer samples with argon (Ar^+^) was carried out at an energy of 500 eV for 30 s to clean the scaffold sample surfaces. The atomic concentration of all relevant elements was calculated from the survey spectra of the samples based on the peak areas of each element (obtained after Shirley background subtraction) divided by the corresponding relative sensitivity factors (RSF). Deconvolution of the XPS spectra was performed accordingly with Shirley background removal, and the peaks were fitted using the Voigt function with an 80% Gaussian and 20% Lorentzian sum.

Raman spectra were measured using an NTEGRA NT-MDT AFM-Raman system (NT-MDT Spectrum Instruments, Moscow, Russia), equipped with a green laser (wavelength: 532 nm) and at a magnification of ×100.

*Wettability.* The water contact angle (WCA) values were measured using the sessile drop method on a drop shape analyzer (DSA 25, KRÜSS, Hamburg, Germany). Three droplets of deionized water with a volume of 2.0 μL were placed at different locations on the surface of each sample. The images were taken after 2 s and after 1 min of water droplets’ contact with the sample surface.

*Surface Morphology and Surface Roughness.* Surface morphology micrographs and root mean square (RMS) roughness values were obtained by the atomic force microscope (AFM, NTEGRA NT-MDT AFM system, NT-MDT Spectrum Instruments, Moscow, Russia) in the semi-contact mode at a scanning area of 40 × 40 µm^2^ of the unmodified and surface-modified PLGA and VDF-TeFE scaffold sample surfaces. All samples were scanned using a monocrystalline silicon cantilever NSG01 (NT-MDT Spectrum Instruments, Moscow, Russia) with a force constant range of 1.45–15.10 N/m and an applied force constant of 5.1 N/m for the measurements carried out.

*Mechanical Properties.* Stress-strain curves, values of the maximum tensile strength and the relative elongation of PLGA and VDF-TeFE scaffolds were evaluated on a tension and compression tester (Instron 3343, Illinois Tool Works, Glenview, IL, USA) with a static load cell of 50.0 N (Instron 2519-102, Illinois Tool Works, Glenview, IL, USA). The traverse speed was set to 10.0 mm/min and the size of the test sample was 10 × 10 mm^2^.

*Thermal Gravimetric Analysis (TGA).* Thermal gravimetric analysis (TGA) of the scaffold samples was performed on a simultaneous thermogravimetric analyzer and differential scanning calorimeter (SDT-Q600, TA Instruments, New Castle, DE, USA). PLGA and VDF-TeFE samples with an area of 1.0 × 1.0 cm^2^ and a weight 1.0–4.0 mg were placed in a heat-resistant pot and heated from 20 °C up to 800 °C at a heating rate of 20 °C/min under an air atmosphere.

*Immunocytochemistry (ICC).* Primary human gingival fibroblasts isolated from healthy donors (Pokrovsky Stem Cell Bank) were placed in a plastic Petri dish and grown until they became confluent (complete cell coverage of the bottom of the Petri dish). Thereafter, the cells were trypsinized (using the enzyme trypsin, purchased from Merck, Darmstadt, Germany) for a better detachment of the cells from the Petri dish) and a suspension of 105 cells (counted by a Neubauer chamber) was applied to the surface of the scaffolds in 24-well plates in each case. The polymer scaffolds prepared with the cells were incubated at 37 °C with 5% CO_2_. After 48 h, the cells attached to the scaffold surfaces were fixed for 20 min with 1% paraformaldehyde (Merck, Darmstadt, Germany), and then for 5 min in methanol (purchased from Ekros, Moscow, Russia) at −20 °C. Subsequently, the scaffolds with the fixed cells were permeabilized in 0.1 *v/w*% Triton X-100/PBS (Triton X-100—nonionic surfactant, purchased from Merck, Darmstadt, Germany; PBS—phosphate buffered saline, purchased from Biolot, Saint-Petersburg, Russia) for 5 min and washed with PBS, followed by a treatment with a blocking solution of 1.0 *w/w*% BSA/PBS (BSA, bovine serum albumin, Merck, Darmstadt, Germany) for 1 h (this step decreases the non-specific binding of the staining antibodies). In order to show the cell morphology on the polymer scaffolds, the cellular cytoskeleton was visualized in the attached cells. Therefore, vimentin and vinculin with the following antibodies were applied: vimentin (V9, sc-6260, Santa Cruz Biotechnology, Dallas, TX, USA) and vinculin (7F9, sc-73614, Santa Cruz Biotechnology, Dallas, TX, USA). Secondary antibodies conjugated with Alexa488 (Invitrogen, Waltham, MA, USA) were used for the final staining. The cell nuclei were visualized with 4′,6-Diamidin-2-phenylindol (DAPI, Invitrogen, Waltham, MA, USA). Fluorescence micrographs were taken at ×20 magnification using an inverse brightfield microscope (Axio Observer 3, Carl Zeiss AG, Oberkochen, Germany).

*Morphology of the Cells on the Scaffold Surfaces.* A scanning electron microscope (SEM, JSM-7001F, Jeol, Akishima, Japan) at ×1200 magnification was used to investigate the morphology of cells on the scaffold surfaces. A thin layer of gold (thickness: 40 nm) was deposited on the scaffolds with cells by magnetron sputtering (Emitech K950 with K350 module, Ashford, Kent, England) to provide charge dissipation while working with the SEM.

*Statistical Data Analysis.* The OriginPro^®^ 2021 software (OriginLab, Northampton, MA, USA) was utilized to determine the significance of differences in the study results. Differences in fiber diameters, pore areas, mechanical properties and wettability were evaluated using the Mann–Whitney U test. The differences are statistically significant with *p* < 0.05.

## 3. Results and Discussion

### 3.1. Physico-Chemical and Mechanical Investigation Results

Macroscopic Appearance of the Scaffolds. Photographs of the upper and lower sides of the unmodified and modified scaffolds are shown in the Appendix A. The u-PLGA and u-VDF-TeFE scaffolds show a white color on the upper and lower sides. Ti-PLGA shows a gray color with a metallic gloss on the upper side. Ti-VDF-TeFE scaffolds show a matte gray color on the upper side. The lower side of the Ti-PLGA and the Ti-VDF-TeFE scaffolds retained their original white color. This indicates that the magnetron sputtering process only modifies the upper layers of the scaffold fibers, while the middle and lower layers remain unmodified.

Surface Morphology. Scanning electron microscope (SEM) micrographs obtained at ×1000 magnification and scaffold fiber diameter and scaffold pore area histograms of the unmodified and surface-modified PLGA and VDF-TeFE scaffolds are shown in Figure 2. Electrospun PLGA and VDF-TeFE scaffolds show the characteristic non-woven structure (Figure 2 to the left). The fiber diameters of PLGA scaffolds are significantly thicker compared to the fiber diameters of VDF-TeFE scaffolds. After the surface modification of scaffolds by titanium, the scaffolds retain their original non-woven structure (Figure 2 to the left). The fiber diameter histograms of the u-PLGA and Ti-PLGA scaffolds display that they consist of ~78 ± 1% fibers with a diameter range of 1.2 to 4 μm. In the case of the VDF-TeFE scaffolds, about 80 ± 3% fibers with a diameter of up to 1 µm are observed. The evaluation of the pore areas shows that ~79 ± 2% of the pores of the unmodified and modified PLGA scaffolds have an area of 10 to 75 μm^2^ and ~81 ± 1% of the pores of the unmodified and modified VDF-TeFE scaffolds have an area of up to 10 μm^2^. Mean fiber diameters are in the range of (1.72–1.80) ± 0.62 μm for all types of PLGA scaffolds and (0.78–0.84) ± 0.34 μm for all types of VDF-TeFE scaffolds (Figure 2 in the middle). The mean pore areas for PLGA and VDF-TeFE samples are in the range of (25.4–27.7) ± 18.2 μm^2^ and (7.2–7.6) ± 6.5 μm^2^, respectively (Figure 2 to the right). In addition, the porosity of all samples is in the range of (56–62) ± 9%. The mean pore area is directly proportional to the fiber diameter of polymer materials obtained by electrospinning [35,36].

The surface-modification of the PLGA and VDF-TeFE scaffolds with titanium does not reliably affect the fiber diameter, pore area and porosity. This indicates that the polymer sample modification in the applied mode makes it possible to preserve their initial surface morphology.

Elemental and Chemical Composition. The energy dispersive X-ray spectroscopy (EDX) spectra of unmodified and surface-modified PLGA and VDF-TeFE scaffolds are shown in Appendix A. In addition to oxygen, carbon, fluorine and titanium, gold is observed in the EDX spectra, which is due to the sputtering of a thin film of gold to create a conductive sample surface. Two peaks can be observed in the EDX spectrum of the unmodified PLGA scaffolds that indicate the presence of carbon (C = 59.9 ± 5.5 at.%) and oxygen (O = 40.1 ± 4.1 at.%) from which the polymer is built (Appendix A). After surface modification, the PLGA scaffolds show, in addition to carbon (C = 53.9 ± 6.7 at.%) and oxygen (O = 43.3 ± 7.8 at.%), two titanium peaks (Ti = 2.8 ± 1.9 at.%) in the EDX spectrum (Appendix A). The EDX spectrum of the unmodified VDF-TeFE scaffolds shows a carbon peak (C = 58.4 ± 5.5 at.%) and a fluorine peak (F = 36.8 ± 3.9 at.%) (Appendix A). After surface modification of the VDF-TeFE scaffolds, a carbon peak (C = 36.8 ± 3.9 at.%), an oxygen peak (O = 20.7 ± 3.9 at.%), a fluorine peak (F = 34.3 ± 4.4 at.%) and two titanium peaks (Ti = 8.2 ± 4.2 at.%) can be observed (Appendix A).

The increase in oxygen concentration after surface modification of PLGA frameworks (Appendix A) as well as the occurrence of oxygen peaks after surface modification of VDF-TeFE frameworks (Appendix A) can be explained as follows:(1)The absorption of oxygen from the atmosphere by the chemically active polymer scaffold surface, which is obtained after a plasma modification [37,38].(2)Under normal conditions, pristine titanium can actively interact with atmospheric oxygen and form an oxide film of ~8 nm thickness [39].

In the case of the surface-modified PLGA and VDF-TeFE scaffolds by titanium, there is a high probability that a metallic oxide film could form on the scaffold surfaces. The plasma modification of non-woven fluoropolymer scaffolds by copper forms a thin metallic film on the scaffold surface, as it was shown in an earlier study [12]. During the magnetron sputtering process, individual atoms and clusters of titanium are deposited on the surface of the polymer fiber and form a thin metallic film. A thin oxide film was most likely formed on the titanium coatings after the magnetron sputtering process due to sample extraction from the vacuum chamber of the magnetron setup and further interaction with the oxygen in the ambient air.

The XPS survey spectra of unmodified and surface-modified PLGA and VDF-TeFE scaffolds before and after ion treatment are shown in Appendix A. After ion treatment of unmodified PLGA scaffolds, the carbon concentration increased 1.3-fold and the oxygen concentration decreased 1.9-fold. For the titanium-modified PLGA scaffolds, the carbon concentration decreased 1.3-fold, the oxygen concentration remained at the same level and the titanium concentration increased 1.2-fold after ion treatment. After ion treatment of unmodified VDF-TeFE scaffolds, the carbon concentration increased 1.2-fold and the fluorine concentration decreased 1.2-fold. For the titanium-modified VDF-TeFE scaffolds, the carbon concentration decreased 1.5-fold, the oxygen content increased 1.1-fold and the titanium content increased 1.2-fold after ion treatment.

The decrease in carbon in surface-modified PLGA and VDF-TeFE scaffolds and the increase in titanium and oxygen after their ion beam treatment (Appendix A, Ti-PLGA, Ti-VDF-TeFE) indicate that a part of the carbon exists directly on the surface of polymer scaffolds. Moreover, this result also indicates a thin film of titanium oxide below the carbon layer. Most likely, this carbon is due to contamination on the surface of the modified samples. The decrease of oxygen in unmodified PLGA scaffolds and the decrease of fluorine in unmodified VDF-TeFE scaffolds after their surface cleaning by plasma treatment (high-energy positively charged particles (Ar+ ion beam; Appendix A, u-PLGA, u-VDF-TeFE) are associated with the removal of scaffold layers close to the surface.

In the Raman spectrum of unmodified PLGA scaffold samples (u-PLGA), the three peaks with high intensity are observed at a Raman shift of 875 cm^−1^, 1453 cm^−1^ and 1770 cm^−1^, indicating stretching of the C–O–C, O–C=O and C=O bonds (Appendix A) [40]. The location and intensity of such peaks are typical for electrospun PLGA scaffolds [40]. After surface modification with titanium, three broad peaks are observed at 269 cm^−1^, 411 cm^−1^ and 600 cm^−1^, indicating the rutile structure of TiO_2_ (Appendix A). The broad shape of the peaks may indicate that an amorphous TiO_2_ thin film has formed on the surface of PLGA scaffolds after their surface modification. In the Raman spectrum of an unmodified VDF-TeFE scaffold, three narrow and high-intensive peaks can be observed, which are located at a Raman shift of 825 cm^−1^, 840 cm^−1^ and 1435 cm^−1^ (Appendix A). The first peak at 825 cm^−1^ is related to some fragments of the polymer [41], and the second peak at 840 cm^−1^ indicates a pronounced polar β-phase, rocking of the CH_2_ bonds and an antisymmetric stretching of the CF_2_ bonds [42]. Finally, the third peak at 1435 cm^−1^ indicates a scissoring and wagging of the CH2 bonds (Appendix A) [42]. After plasma modification of the VDF-TeFE scaffold, the Raman spectrum shows three high-intensity peaks that could be observed at 234 cm^−1^, 448 cm^−1^ and 615 cm^−1^ (Appendix A). The position and intensity of the three peaks in the Ti-VDF-TeFE sample spectrum may indicate that a thin film of TiO_2_ with a rutile crystal structure was formed on the VDF-TeFE surface after modification by magnetron sputtering of titanium (Appendix A) [43]. The C1s, O1s and Ti2p core level spectra of the unmodified and surface-modified PLGA and VDF-TeFE scaffolds are shown in Figure 3.

Based on the results of the survey XPS spectra (Appendix A), it can be concluded that a carbon contamination is present in addition to the polymer-originating part of carbon in the C1s spectra of unmodified and surface-modified PLGA and VDF-TeFE scaffold samples. Three distinct peaks are present in the C1s spectrum of the u-PLGA scaffold. The first peak (marked in red) at an energy value of 284.9 eV indicates the presence of C–C/C–H bonds, and the second and third peaks (marked in blue and green) located at 286.8 and 288.8 eV, indicating the presence of C–O and O–C=O bonds, respectively [44,45,46].

After the deconvolution of the C1s spectra of the Ti-PLGA scaffold, four components can be distinguished. The first three of them are already present in the u-PLGA scaffold spectrum, and the peaks are located at the binding energies of 284.9, 286.6 and 288.8 eV, which correspond to the molecular groups C–C/C–H, C–O and C=O, respectively. After plasma modification, the fourth component is present at a binding energy of 281.8 eV (marked in magenta), which indicates the formation of TiC [47].

The significant decrease in the relative intensity of the C–O, C=O peaks after plasma modification of the PLGA scaffold can be related to their cleavage [48]. As a result, a relevant proportion of the functional groups C–O and C=O are predominantly replaced by Ti, Ti-C and Ti-O compounds in the PLGA polymer backbone (Figure 3). The appearance of Ti-C can also be related to the carbon contamination observed after magnetron sputtering of titanium [27]. Moreover, the significant decrease in C–O and C=O bonds in the spectra of the Ti-PLGA samples may indicate the presence of a carbon contamination, whose peaks (C–C) are significantly more intense than the C–O and C=O peaks.

In the C1s spectrum of the u-VDF-TeFE scaffold, two peaks are present, with maxima at 286.3 and 291 eV that correspond to the molecular groups of CH_2_ (marked in blue) and CF_2_ (red), respectively [49,50]. Spectrum deconvolution shows an additional third peak at 284.6 eV, which corresponds to the molecular groups of C–C and C–H (marked in green) [51]. Deconvolution of the Ti-VDF-TeFE C1s spectrum identifies four molecular groups, three of which are also represented by the u-VDF-TeFE scaffold. These are the molecular groups of CF_2_, CH_2_ and C–C/C–H, which are observed at the binding energies of 290.8, 286.1 and 284.8 eV, respectively. The fourth peak at the binding energy of 288.8 eV (highlighted in pink) corresponds to the functional groups O–C=O/CF_2_–CHF [52].

A significant decrease of the CF_2_ and CH_2_ peaks after plasma modification of the VDF-TeFE scaffold is related to a dehydrofluorination, which consists of a breaking of the molecular bonds: C–F and C–H [53,54]. As a result, fluorine and hydrogen leave the polymer backbone of VDF-TeFE [55]. The C–C/C–H peak increase in Ti-VDF-TeFE testifies to the preservation of the most of the polymer backbone after plasma modification of the VDF-TeFE scaffold. The fourth peak relates to the components O–C=O/CF_2_–CHF in the Ti-VDF-TeFE scaffold, which were formed due to the interaction of free carbon bonds of the polymer backbone with free fluorine and hydrogen atoms obtained during dehydrofluorination, and the oxygen atoms are coming from ambient air. A significant decrease in CF_2_ and CH_2_ bonds in the spectra of the Ti-VDF-TeFE samples may also indicate the presence of carbon contamination, whose peaks (C–C) are significantly more intense than the CF_2_ and CH_2_ peaks.

The O1s spectrum of u-PLGA exhibits one peak with a maximum at a binding energy of 532.2 eV. The deconvolution process revealed two peaks with maxima at the binding energies of 531.9 eV (marked in red) and 533.3 eV (marked in blue). The red peak corresponds to the functional groups C=O/C–OH and the blue peak represents the C–O bond [46,56].

In the O1s spectrum of the Ti-PLGA scaffold, a peak is present at a binding energy of 530.3 eV. A deconvolution of this spectrum reveals three components, two of which are also observed in the u-PLGA scaffold spectrum. The functional groups C=O and C–O are located at the binding energies of 531.8 and 533.1 eV, respectively. A new large peak (marked in green) appears in the O1s spectrum at a binding energy of 530.2 eV after the modification of the PLGA scaffold via titanium and corresponds to TiO_2_ [57,58].

The O1s spectrum of the Ti-VDF-TeFE scaffold exhibits two reflections with peaks at the binding energies of 530.1 and 532.6 eV, and the deconvolution of this spectrum allows distinguishing three other components. The maxima of the red and blue components are at the binding energies of 531.8 and 533 eV, which correspond to C=O/C–OH and C–O, respectively [56,59]. The third component, shown with a green color peak, is the most intense, with a binding energy of 530.1 eV, which corresponds to TiO_2_.

In the O1s spectra of the Ti-PLGA and Ti-VDF-TeFE scaffolds, the large intensity of TiO_2_ peaks facilitates the assumption about the active interaction of titanium with oxygen and the subsequent formation of a titanium oxide film on the surface of the scaffold fibers.

In the Ti2p spectrum of the Ti-PLGA scaffold, a reflex with three peaks is present, which displays maxima of the binding energy at 464.4, 458.8 and 454.6 eV. The peak at the binding energy of 458.8 eV has the highest intensity. Via deconvolution, six distinct peaks can be identified. Red-colored peaks, located at the binding energies of 458.8 and 464.6 eV, correspond to TiO_2_ [60,61]. Blue-colored peaks at the binding energies of 454.3 and 460.5 eV correspond to the chemical components of TiC/TiO [60,62]. Green-colored peaks at the binding energies of 456.5 and 462.3 eV indicate the chemical component of Ti_2_O_3_ [61].

The Ti2p spectrum of the Ti-VDF-TeFE scaffold clearly shows two high-intensity peaks at the binding energies of 458.9 and 464.6 eV. Spectrum deconvolution reveals four distinct peaks. Highly intense peaks (marked in red) at the binding energies of 458.9 and 464.6 eV correspond to TiO_2_. Less intense peaks (marked in blue) at the binding energies of 456.3 and 461.2 eV indicate the presence of TiO/Ti_2_O_3_ compounds.

The appearance of various compounds of titanium oxide (TiO, TiO_2_ and Ti_2_O_3_) may be associated with the interaction of chemically active metal-containing PLGA and VDF-TeFE scaffold surfaces (Ti-PLGA and Ti-VDF-TeFE) with the oxygen of the ambient air [63], which is little present in the sputtering chamber and after sample exposure to the normal atmosphere. Moreover, the presence of such titanium oxide compounds indicates a multilayer titanium oxide film on the surface of the polymer scaffold fibers [39].

*Mechanical Properties.* The values of the tensile stress (Appendix A) and relative elongation (Appendix A) for u-VDF-TeFE and Ti-VDF-TeFE scaffold samples vary within the range of (10.7–12.0) ± 1.8 MPa and (211–234) ± 36%, respectively. For the u-PLGA and Ti-PLGA scaffold samples, the tensile strength and relative elongation are in the range of (3.2–3.6) ± 0.3 MPa and (427–452) ± 65%, respectively (Appendix A).

The surface modification of PLGA and VDF-TeFE scaffold samples with titanium allows for the preservation of their initial mechanical characteristics, which is possible due to carefully chosen working parameters (described in Section 2.2). Therefore, it can be assumed that the titanium coating was deposited on the scaffold surface in a non-destructive manner. Moreover, the surface modification of electrospun materials by plasma treatment, takes place near the surface, while the polymer fibers located in the depth of the material remain intact [12].

Wettability and Roughness. The results of water contact angles (WCA) and roughness of the unmodified and surface-modified PLGA and VDF-TeFE scaffolds are shown in Figure 4. The WCA values after 2 s and after 1 min of interaction with the PLGA and VDF-TeFE scaffold surfaces differ only very slightly and are almost identical. After 2 s of water interaction with the surfaces of unmodified PLGA and VDF-TeFE scaffold samples, their WCA values are 116 ± 3° and 123 ± 3°, respectively (Figure 4a). After 1 min of water interaction with the unmodified PLGA and VDF-TeFE scaffolds, their WCA values were 118 ± 3° and 125 ± 6°, respectively (Figure 4a). According to the WCA values of the unmodified PLGA and VDF-TeFE scaffold samples, it can be clearly seen that they have pronounced hydrophobic properties. After 2 s and after 1 min of contact of the water droplets with the surface of the Ti-PLGA samples, the WCA values for both were found to be 0° in both cases (Figure 4a). The water droplets were partially and completely absorbed by the PLGA scaffold surface within 2 s and 1 min, respectively (Figure 4a). This indicates the water-absorptive and hydrophilic properties of the Ti-PLGA scaffold. The surface of the Ti-VDF-TEFE scaffolds has hydrophilic properties, which are clearly shown by the WCA values of 46 ± 8° and 44 ± 7° after 2 s and 1 min of contact of the water droplets with the sample surfaces (Figure 4a).

A higher wettability of polymer scaffolds surface-modified with titanium is related to the amount of titanium oxide compounds on the polymer fibers. TiO_2_ can lead to the formation of hydroxyl groups (–OH), which contribute to the formation of hydrogen bonds between TiO_2_ on a metal film and water [64]. It is known that titanium oxide-containing thin films prepared by DC magnetron sputtering have hydrophilic properties [65].

Atomic force microscopy micrographs show that the surfaces of unmodified and surface-modified PLGA and VDF-TeFE scaffolds exhibited a non-woven structure (Appendix A), which correlates with the results of scanning electron microscopy (Figure 2). Unmodified and surface-modified PLGA scaffolds (u-PLGA, Ti-PLGA) have a roughness of 836 ± 71 nm and 901 ± 58 nm, respectively (Figure 4b). The roughness of unmodified and surface-modified VDF-TeFE scaffolds (u-VDF-TeFE, Ti-VDF-TeFE) is 714 ± 82 nm and 648 ± 109 nm, respectively (Figure 4b). After plasma modification the roughness of PLGA and VDF-TeFE scaffolds remains unchanged, which correlates with results of fiber diameter and pore size (Figure 2). The roughness of PLGA scaffolds was 1.2–1.4 times higher than the roughness of VDF-TeFE scaffolds (Figure 4b), which is associated with the larger fiber diameter and pore size of PLGA scaffolds (Figure 2).

*The thermal gravimetric analysis.* The thermal gravimetric analysis (TGA) and differential thermal gravimetric analysis (DTG) curves are shown in Appendix A. The initial weights are: 3.22 ± 1.01 mg for the unmodified PLGA scaffolds, 3.77 ± 1.41 mg for the surface-modified PLGA scaffolds, 0.95 ± 0.19 mg for the unmodified VDF-TeFE scaffolds and 1.16 ± 0.23 mg for the surface-modified VDF-TeFE scaffolds (Appendix A). No residual mass is observed after the thermal decomposition of unmodified PLGA and VDF-TeFE scaffolds (Appendix A). Residual masses of titanium coatings are present after thermal decomposition of the surface-modified PLGA (0.29 ± 0.18 mg) and VDF-TeFE scaffold samples (0.09 ± 0.04 mg) (Appendix A). This indicates a titanium content of about 7.5 wt.% for both scaffold sample types (Appendix A). As a result, titanium on surface-modified PLGA scaffolds is 3.5 times higher than on surface-modified VDF-TeFE scaffolds. The thermal decomposition temperatures are found to be as follows: 304 °C for unmodified PLGA scaffolds, 308 °C for surface-modified PLGA scaffolds, 398 °C for unmodified VDF-TeFE scaffolds, and 418 °C for surface-modified VDF-TeFE scaffolds (Appendix A).

Such a significant difference in the mass of titanium on the surfaces of PLGA and VDF-TeFE scaffolds is associated with the significant differences in their surface morphology (Figure 2). The diameter of PLGA fibers is approximately two times larger than the diameter of VDF-TeFE fibers (Figure 2). Since the thin titanium coatings are formed predominantly on the surface of polymer scaffold fibers, it can be expected that a thinner titanium coating is formed on VDF-TeFE fibers than on PLGA fibers. During the coating formation, individual titanium atoms are more actively reflected from the surface of VDF-TeFE fibers since such fibers are thinner and have a lower roughness compared to the fiber size (Figure 2) and roughness (Figure 4b) of PLGA scaffolds.

### 3.2. Immunocytochemistry and Morphology of Living Cells on the Scaffolds

The SEM micrographs and immunocytochemical staining of human gingival fibroblast cells placed on the unmodified and surface-modified PLGA and VDF-TeFE scaffold surfaces are displayed in Figure 5. As seen in the SEM micrographs (Figure 5, left column), human gingival fibroblast cells spread widely and form an extension on the surface of the u-PLGA, Ti-PLGA and Ti-VDF-TeFE scaffolds, where the morphology of the cells on the u-VDF-TeFE scaffolds is abnormal. Fewer human gingival fibroblasts are observed on SEM micrographs (Figure 5, left column) of the u-VDF-TeFE scaffold samples, as confirmed by optical fluorescence micrographs (Figure 5, middle column). As can be seen in the figures for u-VDF-TeFE samples, the fibroblasts (Figure 5, u-VDF-TeFE) are much less ordered than fibroblasts on the surfaces of other samples (Figure 5, u-PLGA, Ti-PLGA, Ti-VDF-TeFE). The cells on the u-PLGA, Ti-PLGA and Ti-VDF-TeFE scaffolds have a more defined fibroblast-like morphology and show a more distinct cell colonization on the scaffolds with a surface modification. Titanium modification dramatically improves the cellular adhesion for both types of polymer scaffolds (made of PLGA and VDF-TeFE). Moreover, the Ti-PLGA scaffold in particular shows the highest level of cell colonization (Figure 6), while the WCA for this type of scaffold is the lowest (Figure 4a), which in general promotes the cell adhesion.

The number of gingival fibroblast cells per area on unmodified and surface-modified PLGA and VDF-TeFE scaffolds is shown in Figure 6. As it can be seen, the immunocytochemical analysis via fluorescence micrographs shows that the number of adherent cells after 48 h of cultivation on the PLGA scaffold samples is significantly higher than on the VDF-TeFE scaffold samples (Figure 6). Fibroblasts did not form confluent surface coverage and therefore did not show proper cell settlement on VDF-TeFE scaffolds. In contrast, the PLGA scaffolds are better suited for cell adhesion. The surface modification by titanium of the PLGA and VDF-TeFE scaffolds results in a significant increase in the cell numbers compared to the unmodified counterparts of these scaffolds. Forty-eight hours after seeding, the fibroblasts grown on the Ti-PLGA surface showed good spreading, and additionally, organized actin filaments were formed regularly. This means that the surface modification of polymer scaffolds via titanium significantly improves cell adhesion and proliferation.

The better cell adhesion and proliferation on PLGA than on VDF-TeFE scaffolds is related to higher values of the pore area (Figure 2). Values for the pore area of PLGA scaffolds are ~4 times higher than for the VDF-TeFE scaffolds (Figure 2). An increase in the pore size (up to ~150–300 microns) of polymer scaffolds has a positive effect on cell proliferation and adhesion [66]. PLGA scaffolds have fibers ranging from nanometers to micrometers in diameter. Such a combined nano- and micro-fiber structure of polymer scaffolds fabricated by electrospinning shows a higher biocompatibility compared to scaffolds that consist only of fibers in the nanometer or micrometer range [67]. The values for the mean fiber diameter and the pore area of VDF-TeFE scaffolds (Figure 2) are not optimal for the proliferation of human gingival cells. Chen et al. concluded that electrospun polymer scaffolds with fiber diameters of 800–900 nm exhibit a low cell proliferation rate [68], which is also in the range of the fiber diameters of 0 to 1600 nm fabricated and investigated here. Moreover, small scaffold pores reduce the efficiency of cell settlement and thus prevent the integration of cells into the scaffold volume structure [69]. Unmodified PLGA and VDF-TeFE scaffolds show low proliferative activity compared to the corresponding surface-modified scaffold samples. This is primarily due to the low wettability (high WCA/low hydrophilicity) of unmodified scaffolds (Figure 4a). A low wettability of polymeric materials limits the proliferative activity of cells [70]. A significant improvement in the cell adhesion and proliferation properties of the surface-modified PLGA and VDF-TeFE scaffolds is associated with an increase in their wettability and the presence of hydrophilic surface properties. In general, the plasma modification of polymer materials can favorably influence their wettability and biocompatibility [71].

## 4. Conclusions

In this work, poly(lactic-co-glycolic acid) and the copolymer vinylidene fluoride-tetrafluoroethylene barrier scaffolds were successfully fabricated by electrospinning and surface-modified by titanium applied via pulsed direct current magnetron sputtering in an argon atmosphere. This modification process forms a thin metallic film on the surface of the scaffolds. The applied deposition parameters were selected to avoid damaging the scaffold fibers so that the morphology and mechanical properties of the scaffolds were preserved. The wettability of the polymer scaffolds increases after the surface modification, and the scaffold surfaces become more hydrophilic, which benefits cell adhesion. During the magnetron sputtering process, the vinylidene fluoride-tetrafluoroethylene scaffold samples undergo a significant dehydrofluorenation on the surface fibers, where carbon-oxygen bonds are particularly replacing the fluorine compounds. The coatings consist of titanium oxide compounds (titanium(II) oxide, titanium dioxide and titanium(III) oxide), which were formed through the surface modification process and can also be associated with oxygen in the sputtering chamber, which is present in small amounts and after the sample has been exposed to the normal atmosphere. In addition, the thermal gravimetric analysis results show a more pronounced titanium coating for the poly(lactic-co-glycolic acid) scaffold samples, both by weight and weight percent. The investigation of the human gingival fibroblast colonization displays higher values for the surface-modified scaffolds compared to the unmodified scaffolds. After the magnetron sputtering of titanium, the highest cell adhesion was found for the surface-modified poly(lactic-co-glycolic acid) scaffolds, which can be recommended for applications in the field of periodontal tissue regeneration.

## Figures and Tables

**Figure 1 polymers-14-04922-f001:**
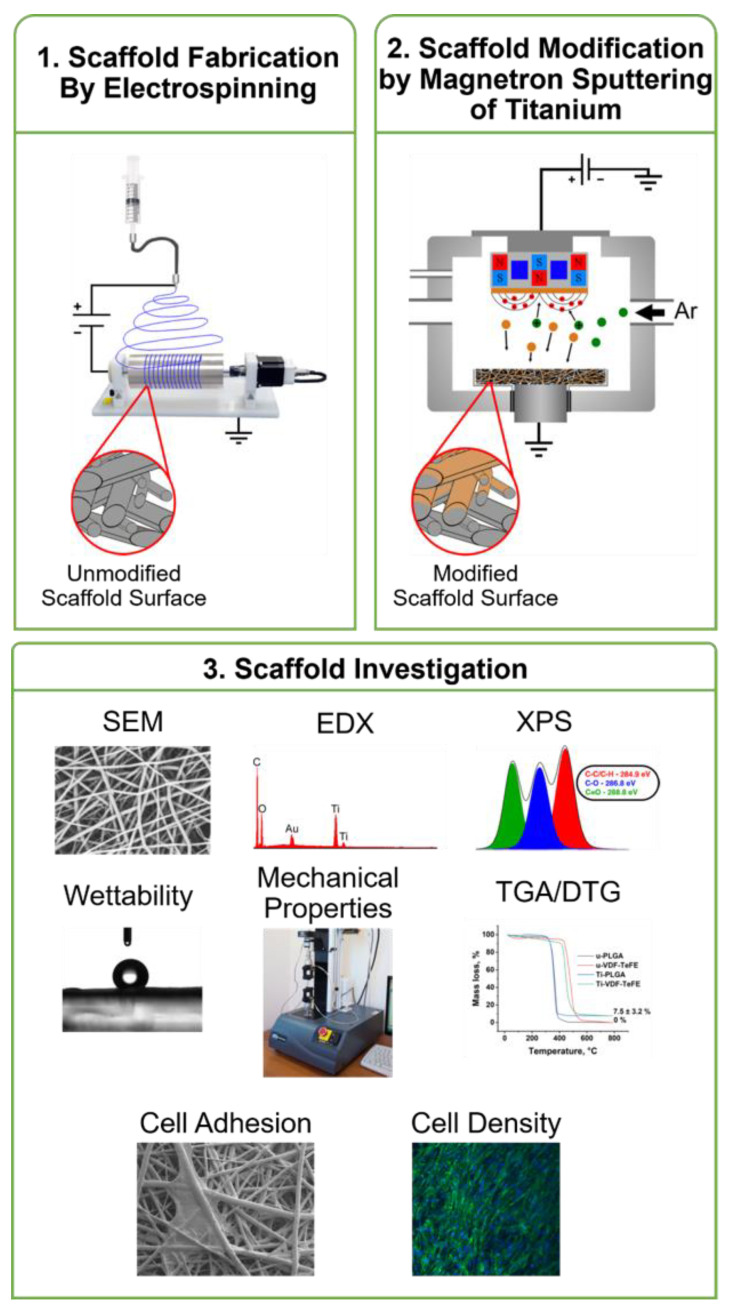
Experimental scheme of (**1**) the fabrication of the poly(lactide-co-glycolide) (PLGA) and vinylidene fluoride-tetrafluoroethylene (VDF-TeFE) copolymer scaffolds via electrospinning; (**2**) the subsequent surface modification by pulsed DC magnetron sputtering of titanium; and (**3**) the investigation of the mechanical properties of all scaffold samples and the investigation of the adhesion of gingival fibroblasts to the scaffold sample surfaces.

**Figure 2 polymers-14-04922-f002:**
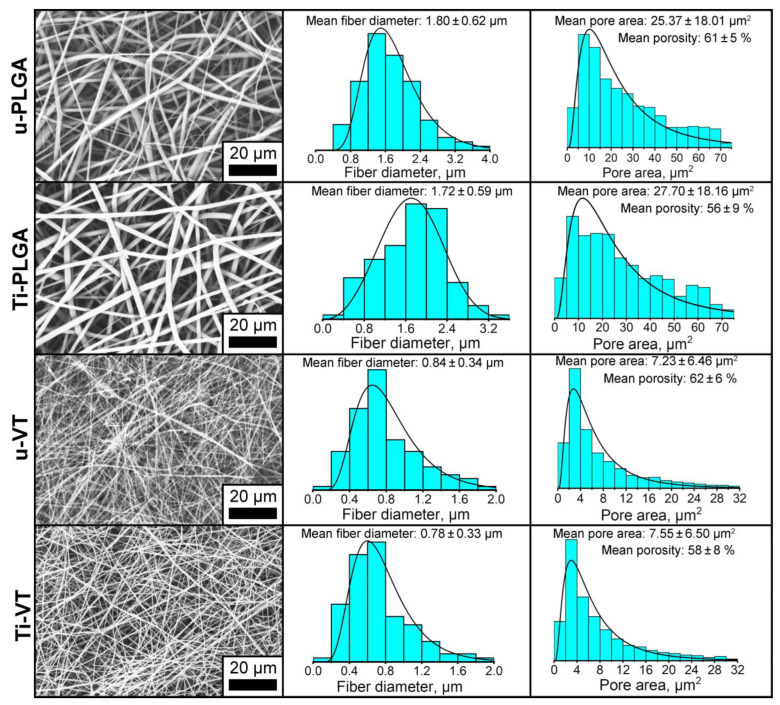
Scanning electron microscope (SEM) micrographs on the left side, histograms of scaffold fiber diameter in the middle and histograms of the scaffold pore area on the right side of u-PLGA, u-VDF-TeFE, Ti-PLGA and Ti-VDF-TeFE scaffolds.

**Figure 3 polymers-14-04922-f003:**
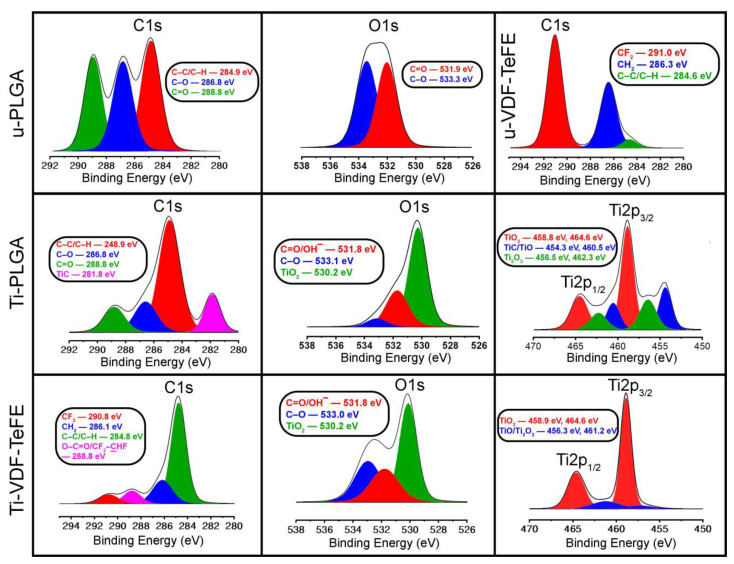
The C1s, O1s and Ti2p XPS spectra of u-PLGA, u-VDF-TeFE, Ti-PLGA and Ti-VDF-TeFE scaffolds. The u-PLGA and u-VDF-TeFE scaffold samples have no corresponding XPS spectra due to the lack of some elements in the polymer surfaces (O1s for the u-VDF-TeFE scaffold and Ti2p for both unmodified scaffolds).

**Figure 4 polymers-14-04922-f004:**
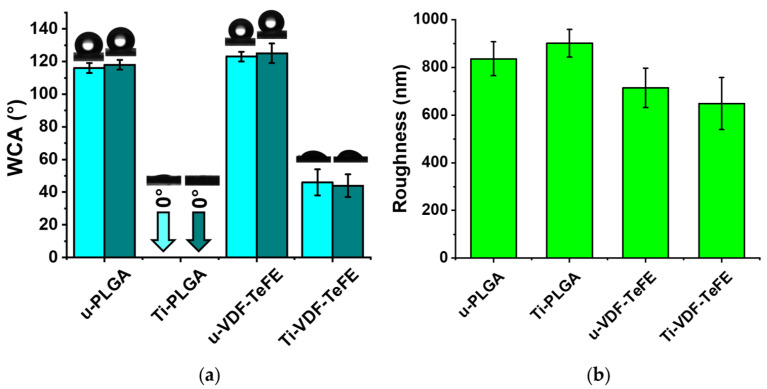
Water contact angles (WCA) and roughness of u-PLGA, Ti-PLGA, u-VDF-TeFE and Ti-VDF-TeFE scaffolds: (**a**) WCA values after 2 s of liquid contact with the scaffold’s surface (marked in cyan) and after 1 min (marked in dark cyan); (**b**) roughness values (marked in light green). The WCA of Ti-PLGA scaffolds is shown as 0°. All values are shown as mean ± SD.

**Figure 5 polymers-14-04922-f005:**
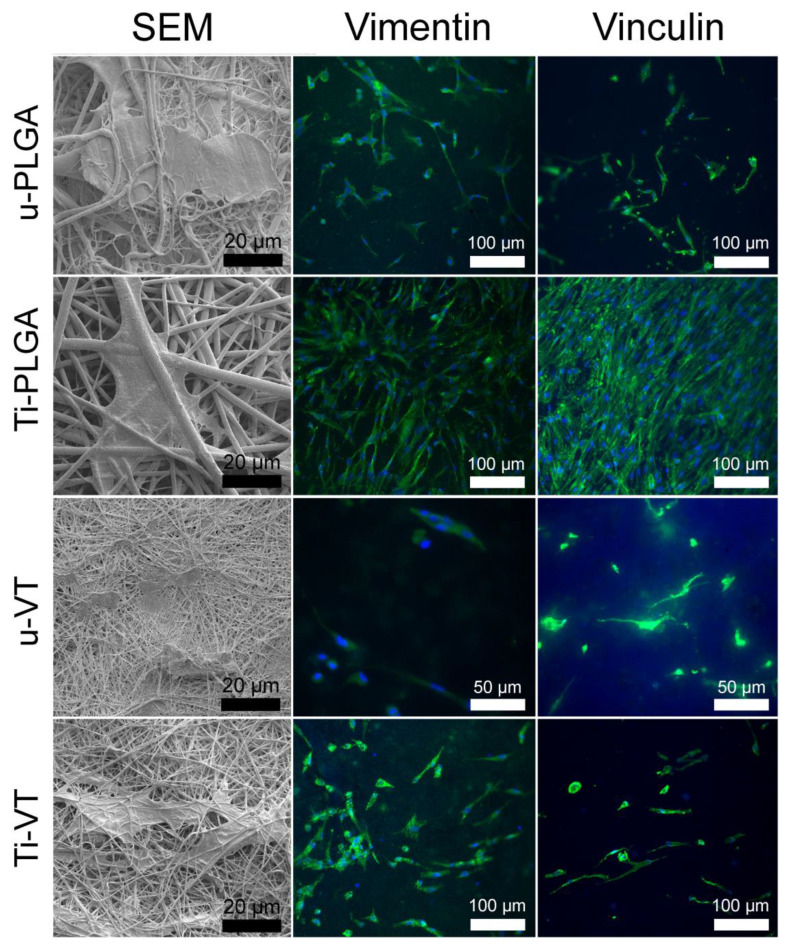
SEM micrographs of human gingival fibroblasts (on the **left**) and optical fluorescence micrographs of u-PLGA, u-VDF-TeFE, Ti-PLGA and Ti-VDF-TeFE after 48 h of cell cultivation. Under fluorescent light, the nuclei of the cells appear blue due to staining with DAPI, and the cytoplasm of the cells appears green due to antibody labeling with vimentin (in the **middle**) and vinculin (on the **right**) and staining with fluorescence dye Alexa488 (see Section 2.2 for more details).

**Figure 6 polymers-14-04922-f006:**
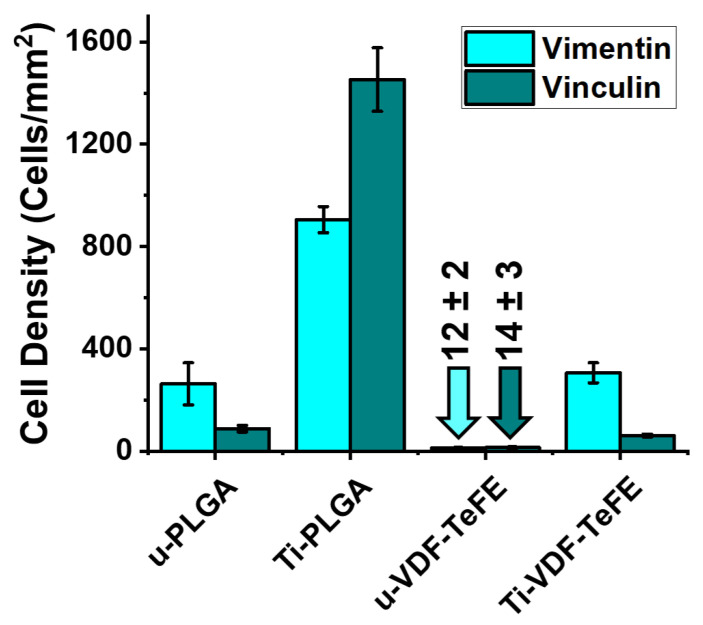
Number of cells per area (cells/mm^2^) on u-PLGA, u-VDF-TeFE, Ti-PLGA and Ti-VDF-TeFE scaffold samples for both types of both utilized antibodies to labelling the primary human gingival fibroblasts with Alexa488 for fluorescence staining (vimentin—cyan, vinculin—teal).

## Data Availability

Data underlying the results presented in this paper are not publicly available at this time but may be obtained from the authors upon reasonable request.

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
