# Peer review of "Surface Modification of Electrospun Bioresorbable and Biostable Scaffolds by Pulsed DC Magnetron Sputtering of Titanium for Gingival Tissue Regeneration"

_polymers, 2022, doi:10.3390/polym14224922_

Round 1

Reviewer 1 Report

The study design of this article is appropriate and useful in expanding our understanding of the issue. Well-written title, abstract, and introduction are presented in this study.  Literature about previous studies is presented in sufficient detail for readers to understand the rationale and significance of the present study. In the materials and methods section, each approach used in the study is thoroughly detailed, well written, and appropriate. There is, however, some aspects of the articles that could be improved:

1. In line 96, the authors mention using hexafluoroisopropanol to make PLGA solution. Wouldn't it be better to use a less toxic solvent? And how did the authors ensure that solvents eventually evaporated?

2.  I’m wondering way the authors decided to investigate VDF-TeFE in the first place as it’s a Non-degradable and hydrophobic polymer.

3. In discussion section, line 576, the authors state “The better cells adhesion and proliferation on PLGA than on VDF-TeFE scaffolds is related with higher values of the pore area”. It’s better to also discuss the relation between “cell adhesion and proliferation” and the “hydrophilicity” of unmodified PLGA and VDF-TeFE scaffolds.

Author Response

The study design of this article is appropriate and useful in expanding our understanding of the issue. Well-written title, abstract, and introduction are presented in this study.  Literature about previous studies is presented in sufficient detail for readers to understand the rationale and significance of the present study. In the materials and methods section, each approach used in the study is thoroughly detailed, well written, and appropriate. There is, however, some aspects of the articles that could be improved:

Answer: We would like to thank the reviewer for his helpful comments and judgement.

  1. In line 96, the authors mention using hexafluoroisopropanol to make PLGA solution. Wouldn't it be better to use a less toxic solvent? And how did the authors ensure that solvents eventually evaporated?

Answer: We thank the reviewer for this good question. Yes, it would be very good to use less toxic solvents. In our preliminary tests, we found hexafluoroisopropanol to be a good solvent for the PLGA used. Another point to take into account is the electrospinning behavior of the dissolved PLGA. We also tried chloroform where the resulting electrospun scaffolds were of minor quality. In order to remove the excess solvent, the samples were dried for 48 h in a drying oven at room temperature and a pressure of 0.5 Pa. This information has been added to chapter 2.1.

  1.  I’m wondering way the authors decided to investigate VDF-TeFE in the first place as it’s a Non-degradable and hydrophobic polymer.

Answer: We thank the reviewer for this remark. The field of application of current polymeric scaffolds is the regeneration of the soft tissues of the oral cavity. Therefore, there is no need to use only biodegradable polymer since it is also possible to easily remove the polymer scaffold from the site of implantation in a patient's gums. VDF-TeFE is a very promising material for biomedical applications where polymer biodegradability is not required. The promising use of VDF-TeFE scaffolds has been well studied in our research group and we were able to show that VDF-TeFE can also be used to regenerate: injured skin [10.3390/membranes11010021] and soft tissue wounds of the oral cavity [10.1016/j. apsusc.2019.144068] can be used. The idea of this study is therefore to directly compare PLGA and VDF-TeFE regeneration properties of gingivual tissue. For this purpose, both polymer scaffolds were treated with the same surface modification method. In different studies, we try to compare common polymers for tissue engineering with the copolymer VDF-TeFE, because this copolymer must be investigated much more for biomedical applications.

  1. In discussion section, line 576, the authors state “The better cells adhesion and proliferation on PLGA than on VDF-TeFE scaffolds is related with higher values of the pore area”. It’s better to also discuss the relation between “cell adhesion and proliferation” and the “hydrophilicity” of unmodified PLGA and VDF-TeFE scaffolds.

Answer: We thank the reviewer very much for this helpful comment. A discussion on this point was added in the last paragraph before chapter 4.

Reviewer 2 Report

The authors reported an interesting work. Four different types of scaffolds were fabricated and characterized. The reviewer has the following comments/suggestions:

1. Title: The word "better" could be removed as the current work doesn't directly investigate the superiority of the fabrication scaffolds by comparing them to other systems.

2. Avoid single-sentence paragraphs. And the manuscript should be carefully proofread.

3. Some of the subtitles could be revised. For example, the title for section 2.3 can be Characterization of Scaffolds.

4. Figures 4 and 6: remove the numbers from the graphs' columns that are unnecessary.

5. Avoid using the term "biocompatibility." The data provided doesn't fully assess the board aspects of biocompatibility.

Author Response

The authors reported an interesting work. Four different types of scaffolds were fabricated and characterized. The reviewer has the following comments/suggestions:

Answer: We would like to thank the reviewer for his helpful remarks.

  1. Title: The word "better" could be removed as the current work doesn't directly investigate the superiority of the fabrication scaffolds by comparing them to other systems.

Answer: We thank the reviewer for this suggestion. The title of the manuscript has been adjusted.

  1. Avoid single-sentence paragraphs. And the manuscript should be carefully proofread.

Answer: We are thankful to the reviewer for pointing out such issues. The paragraphs have been restructured according to the comment during our proofread.

  1. Some of the subtitles could be revised. For example, the title for section 2.3 can be Characterization of Scaffolds.

Answer: We thank the reviewer for this helpful comment. All section titles has been revised and adjusted to the common MDPI style.

  1. Figures 4 and 6: remove the numbers from the graphs' columns that are unnecessary.

Answer: We thank the reviewer for this comment. We like this kind of information in a chart because we think it gives all the information at a glance and we find it more convenient for the readers. Nevertheless, we have adjusted Figures 4 and 6 accordingly. We have only left the information that we consider necessary.

  1. Avoid using the term "biocompatibility." The data provided doesn't fully assess the board aspects of biocompatibility.s

Answer: We thank the reviewer for this remark. Unfortunately, we cannot agree with this comment completely. In some places in the manuscript text with citations, we also refer to other studies related with this term. In these cases, we have kept it. For all other cases, we have removed or replaced this term. According to the definition, biocompatibility is given when materials or assemblies have no negative impact on living beings in their environment. The data presented here does not show any negative influence, which means that we refer to the term “biocompatibility”. Both copolymers are generally classified as biocompatible while exhibiting different properties. Since PLGA is considered to be biodegradable, we wanted to improve cell adhesion, which the results also show. VDF-TeFE is also known to be repellent to cell adhesion, so the question was whether a Ti coating could increase the cell friendliness of this biostable copolymer. But that's not a problem either, because there are areas where cells shouldn't settle on implants, or only to a very limited extent. Like artificial blood vessels, for example. Our working group is already using the results of this study in in-vivo studies, which have not been published yet.
